# Metabolomics Research in Periodontal Disease by Mass Spectrometry

**DOI:** 10.3390/molecules27092864

**Published:** 2022-04-30

**Authors:** Sachio Tsuchida, Tomohiro Nakayama

**Affiliations:** Division of Laboratory Medicine, Department of Pathology and Microbiology, Nihon University School of Medicine, 30-1 Ooyaguchi-kamimachi, Itabashi-ku, Tokyo 173-8610, Japan; tsuchida.sachio@nihon-u.ac.jp

**Keywords:** periodontitis, periodontal disease, metabolomics research, proteomic analysis, LC-MS/MS, GC-MS

## Abstract

Periodontology is a newer field relative to other areas of dentistry. Remarkable progress has been made in recent years in periodontology in terms of both research and clinical applications, with researchers worldwide now focusing on periodontology. With recent advances in mass spectrometry technology, metabolomics research is now widely conducted in various research fields. Metabolomics, which is also termed metabolomic analysis, is a technology that enables the comprehensive analysis of small-molecule metabolites in living organisms. With the development of metabolite analysis, methods using gas chromatography–mass spectrometry, liquid chromatography–mass spectrometry, capillary electrophoresis–mass spectrometry, etc. have progressed, making it possible to analyze a wider range of metabolites and to detect metabolites at lower concentrations. Metabolomics is widely used for research in the food, plant, microbial, and medical fields. This paper provides an introduction to metabolomic analysis and a review of the increasing applications of metabolomic analysis in periodontal disease research using mass spectrometry technology.

## 1. Introduction

Periodontal disease includes any disorder occurring in periodontal tissue, consisting of the gingiva, cementum, periodontal ligament, and alveolar bone (Figure 1) [1,2,3,4,5,6,7]. The main periodontal diseases are gingival lesions and periodontitis, but also include nonplaque gingival lesions, gingival proliferation, necrotizing periodontal disease, abscesses of periodontal tissue, periodontal and endodontic lesions, gingival recession, and occlusal trauma [8,9,10,11,12,13,14]. Periodontal disease is one of the two major dental diseases, along with dental caries, and is the most significant cause of tooth loss. Recently, it has also become clear that periodontal disease is associated with systemic diseases, such as diabetes and aspiration pneumonia, and is now addressed as a lifestyle-related disease.

Recently, metabolome analysis has made rapid progress. It is an analytical method that comprehensively searches for metabolites in cells and biological samples to elucidate the interaction between biomacromolecules and metabolites, novel metabolic pathways, unknown metabolic regulatory mechanisms, and unknown gene and protein functions. It has become widespread. The contribution of mass spectrometry is also considered significant. Mass spectrometry and metabolome analysis are indispensable in basic research on periodontal diseases, and are now being applied to periodontal diseases. Metabolomics is a tool of personalized medicine and is applicable to several diseases, from autoimmune to chronic diseases, and lately it has found utility in diagnosing and treating periodontal disease. The correct diagnosis and treatment of periodontal disease are crucial for preventing tooth loss and improving the patient’s quality of life. Metabolomics, also known as metabolomic analysis, is a technology used to comprehensively analyze small-molecule metabolites in living organisms [15,16].

Mass spectrometry is a method for identifying (finding out what something is) or quantifying (measuring the amount of) a substance by measuring its mass and number of ions, which are made into minute ions at the atomic and molecular levels using various ionization methods. With the development and improvement of metabolite analytical methods using mass spectrometry (MS), such as gas chromatography–mass spectrometry (GC-MS) and liquid chromatography–mass spectrometry (LC-MS), it has become possible to analyze various metabolites and to detect metabolites at lower concentrations [17,18,19,20].

MS introduction in clinical settings is expected to increase further, enabling the continuous collection of data on periodontal diseases while researchers strive to update technology and accumulate more information. It is expected that the use of metabolomics will contribute toward improving research outcomes and the treatment of periodontal diseases. Recently, metabolomics has been applied in periodontology. This review introduces metabolomics research and provides an overview of its increasing applications in periodontal disease using MS technology.

## 2. Periodontal Disease

Periodontology is a new field relative to other areas of dentistry. In recent years, there has been remarkable progress in research, clinical practice, and education related to periodontology [21,22,23]. As suggested by the term “periodontal medicine,” it is associated with systemic diseases. Metabolic disease groups that have been reported to be associated with periodontal disease include diabetes, obesity, nonalcoholic steatohepatitis (NASH), dyslipidemia, and osteoporosis (Figure 2). Therefore, overcoming periodontal disease through appropriate treatment can help preserve teeth and contribute significantly to improving patients’ quality of life.

A periodontal examination, which is also called a periodontal histology examination, is used to determine the progress and causes of periodontal disease and to formulate a diagnosis and treatment plan. It is also used to evaluate the response of the periodontal tissues and modify the treatment plan accordingly. The examination includes the examination of gingival inflammation, pocket depth, attachment level, oral hygiene, and root bifurcation lesions [24,25,26]. Furthermore, a periodontal precision examination includes the pocket measurement of at least six points per tooth, measurement of tooth mobility, and measurement of plaque adhesion using a plaque chart [27,28,29,30].

Bacteria are involved in the development or progression of periodontitis and are often found at active sites of periodontitis [31,32,33,34,35]. Generally, Koch’s principle must be considered for an infection to be considered to have occurred from a specific pathogenic bacterium [36]. However, since this principle was initially intended for severe infections and is difficult to apply to periodontal disease, Socransky et al. reported a modified version of this principle as a criterion for identifying the source of periodontal disease [37]. Subsequently, in 1996, the American Academy of Periodontology summarized the conditions of periodontopathogenic bacteria and classified most of them as a red or orange complex, except for *A. actionmycetemcomitans*, and bacteria belonging to the red complex are now classified as severely periodontal [38]. These bacteria are associated with inflammation. Periodontopathic bacteria are Gram-negative rods that essentially possess endotoxin as a virulence factor. All bacteria belonging to the red complex metabolize trypsin-like enzymes. The elimination of these bacteria can stop periodontitis progression. Bacteria such as *Porphyromonas gingivitis*, *Tannerella forsythia*, and *Treponema denticola* are commonly detected in chronic periodontitis, whereas *Aggregatibacter actinomycetemcomitans* and others are often detected in invasive or juvenile periodontitis [39,40,41,42]. Hundreds of bacteria present in the oral cavity have been classified in relation to their relevance to periodontal disease [37,43,44,45]. *Tannerella forsythia* (P,g, T,d, and T,f) are believed to have the largest influence on severe periodontitis [46,47,48].

## 3. Mass Spectrometry of Metabolomics Analysis

In recent years, metabolomic analysis has become widely used and can indicate the presence of various factors that influence the results of metabolite analysis related to the processes of sample collection before metabolite analysis, sample pretreatment, and post-analysis data processing [16,49,50,51].

With recent advances in MS technology, metabolomics research is now widely conducted in various research fields [52,53,54]. Recent advances in MS technology have resulted in discovering novel markers and diagnostic biomarker candidates for various diseases. MS-based diagnostic tests could thus be clinically implemented in the future. Areas where MS technology is used in clinical laboratories are listed in Table 1 by procedure, indicating considerable progress [55,56,57,58,59,60,61,62,63,64,65,66,67,68,69,70,71,72,73,74,75,76,77,78,79,80,81,82,83,84,85,86,87,88]. We indicated the characteristics of typical mass spectrometric methods, MALDI-TOF MS, LC-MS, and GC-MS, in Table 1. As the practical applications of matrix-assisted laser desorption/ionization MS have progressed, the combination of electrospray ionization and LC-MS/MS has rapidly emerged as a popular method for determining low molecular weight compounds.

Developments in metabolite analysis methods using GC-MS, LC-MS, and capillary electrophoresis-MS have made it possible to analyze various metabolites and to detect metabolites at lower concentrations [89,90,91,92]. Metabolomics has been extensively used for research in the food, plant, microbial, and medical fields. Since in vivo metabolites, which are the targets of metabolomics, are located downstream of the central dogma that operates biological activities, quantifying in vivo metabolites can provide a more detailed understanding of cellular functions [93,94]. Furthermore, changes in in vivo metabolites are thought to closely reflect changes in the phenotype of an organism [95,96].

Depending on the metabolomics approach, metabolite analysis is conducted on the obtained extracts using various analytical techniques [97,98]. Among the various techniques, MS is currently the mainstay of metabolomics analysis, as it is the most sensitive method and can obtain more information from a smaller sample volume [57,99]. To perform a comprehensive analysis of metabolites, it is important to analyze the maximum possible number of metabolites simultaneously. A wide range of metabolites can be analyzed by combining several methods of MS analysis to compensate for the limitations of each approach [17,100,101,102]. GC-MS is the most common combined form of MS analysis. Recently, there has been increasing interest in using two-dimensional GC-MS to perform a more sensitive analysis [103,104,105]. In addition to GC-MS, LC-MS, which is a combination of high-performance liquid chromatography and MS, is widely used in metabolomics [57,106,107]. 

## 4. Applications of Metabolomics Research in Periodontal Disease Using MS

Metabolomics can be made more effective by combining omics evaluation (genomics and proteomics) with other MS technologies. There are several instances of MS and related techniques being used in the metabolomic analysis of periodontal disease (Table 2 and Figure 3). 

Kuboniwa et al. examined the relationship between salivary metabolites to reflect periodontal inflammation severity using a recently proposed parameter (PISA) and salivary metabolic profiles; metabolic profiling of saliva was performed using gas chromatography coupled with time-of-flight MS, followed by multivariate regression analysis with orthogonal projections to latent structures (OPLS) [108]. Based on the variable importance in the projection values obtained via OPLS, eight metabolites were identified as potential indicators of periodontal inflammation, of which the combination of cadaverine, 5-oxoproline, and histidine yielded a satisfactory accuracy for periodontitis diagnosis [108]. In particular, the suggested involvement of 5-oxoproline (pyroglutamic acid) in periodontal disease is considered interesting. Pyroglutamic acid is an amino acid in which the carboxyl and amino groups of glutamic acid undergo an intramolecular condensation reaction to form a lactam. Increased pyroglutamic acid (5-oxoproline) excretion suggests an abnormality in the metabolic pathway involved in response to oxidative stress and the synthesis of the intracellular reducing agent, glutathione.

Liebsch et al. described the clinical attachment level, periodontal probing depth, supragingival plaque, supragingival calculus, number of missing teeth, and removable denture as oral parameters using a large set (284) of salivary metabolites obtained by LC-MS/MS from a subsample of 909 nondiabetic participants from the Study of Health in Pomerania [109]. The metabolites associated with periodontal disease were suggested to be related to tissue destruction, host defense mechanisms, and bacterial metabolism, with the bacterial metabolite, phenylacetae, being significantly associated with periodontal disease variables [109]. Thus, bacterial metabolites, such as phenylacetate, are expected to be deeply involved in periodontal disease. In another study, Huang et al. combined mass spectrometry (ICP–MS system, GC-MS, LC-MS)-based ionomics and targeted lipidomics on fatty acid metabolites [110]. Ionomics identified decreased salivary levels of Mn, Cu, and Zn in periodontal patients. SOD levels were reduced in saliva and serum in the periodontal group. Elevated levels of cyclooxygenase (COX) products (PGE2, PGD2, and PGF2α and TXB2) in periodontal saliva indicate an enhanced inflammatory response. An increased level of lipoxygenase (LOX) products 5-Hydroxyeicosatetraenoic acid (5-HETE). The oxidative stress marker F_2_-isoprostane was significantly increased in periodontal saliva [110]. Cyclooxygenase (COX) is a functional molecule that influences inflammatory responses’ induction in vivo [111,112]. Lipoxygenases (LOD) are of interest in inflammatory diseases, such as atherosclerosis [113,114]. It is now known that molecules altered by oxidative stress accumulate in various diseases. For example, in diabetes mellitus, oxidized sugars bind to proteins, increasing abnormal glycated proteins. Therefore, research on COX products, LOX products’ oxidative stress, and their involvement in periodontal disease is expected to continue gaining attention.

Ozaki et al. investigated the usefulness of GC-MS, which could be used for the onsite analysis of metabolites in gingival crevicular fluid (GCF), to objectively diagnose periodontitis at the molecular level [115]. GCF is the proximal fluid closest to the lesion site, and best reflects the condition of periodontal tissue (Figure 1). GCF contains many enzymes and proteins related to periodontal tissue metabolism and is considered a significant indicator of its progression [116]. Although the number of enzymes and proteins in GCF is minimal, MS can be used to analyze these trace amounts [117,118,119]. GCF is expected to contain candidate periodontal disease markers that could be assessed by metabolomics analysis. Thus, metabolomics is considered a crucial approach for understanding GCF. In a study by Ozaki et al. using GCF and GC-MS, the peak areas of putrescine, lysine, and phenylalanine were significantly higher in the group of deep-pocket sites than in healthy and moderate sites [115]. Furthermore, ribose, taurine, 5-aminovaleric acid, and galactose were significantly higher in the group of deep-pocket sites compared to healthy and moderate-pocket sites. Lactic acid, benzoic acid, glycine, malic acid, and phosphate gradually increased from healthy to moderate-pocket to deep-pocket sites [115]. GCF successfully detected several metabolites using GC-MS. 

Recently, Chen et al. characterized the gingival metabolome of mice with high-fat diet (HFD)-induced obesity with and without periodontitis [120]. The gingival metabolome and arginine metabolism were analyzed by nontargeted/targeted LC-MS. They concluded that the obese population fed an excessive HFD displayed an amplified metabolic response to periodontitis, exhibiting metabolic susceptibility to exacerbated periodontal destruction [120]. This study indicated the use of gingival metabolome based on MS technology nontargeted/targeted LC-MS in periodontal disease research and potential applications.

Furthermore, it can determine the total number of caries and bacteria related to periodontal disease. Focusing on saliva, Schulte et al. developed three complementary LC-MS/MS approaches, namely targeted multiple reaction monitoring (MRM) LC-MS/MS, nontargeted quantification by data-independent acquisition (DIA, SWATH), and identification and relative quantification of unknown metabolites related to HIV infection and periodontitis by data-dependent acquisition [121]. The PHIV discovery-based dataset identified 564 endogenous peptides in which proteolytic processes and amino acid metabolism occurred [121]. The association with HIV infection was examined for the first time. The salivary metabolite profile was rich in cadaverine, a metabolite known to be associated with periodontitis. It is known that the odoriferous gases in halitosis are volatile amine gases (volatile nitrogen compounds) caused by cadaverine and putrescine [122]. Therefore, it has a significant effect on halitosis in periodontal disease. Further analysis is expected to clarify the mechanisms of periodontal disease and HIV infection.

Overmyer et al. conducted 16S rDNA sequencing as well as metabolomics, lipidomics, and proteomics analyses, including GC-MS and LC-MS/MS, on supragingival dental plaque collected from individuals with prediabetes and type 2 diabetes (Pre-DM/DM), Pre-DM/DM and periodontal disease (PD), PD alone, or neither [123]. Phosphatidylcholines, plasmenyl phosphatidylcholines, ceramides containing non-OH fatty acids, and host proteins related to actin filament rearrangement were elevated in plaques from PD versus non-PD samples [123]. The strong association between Lautropia and monomethyl phosphatidylethanolamine (PE-NMe) is striking because PE-NMe synthesis is uncommon in oral bacteria [123]. Using omics and 16S rDNA sequencing, a novel microbial metabolic pathway and significant associations of host-derived proteins with PD were observed. It is known that people with diabetes have a high incidence of periodontal disease, and diabetes has been viewed as a risk factor for periodontal disease [124,125,126]. Research has also been conducted on the effect of periodontal disease on glycemic control with diabetes itself, and the bidirectional relationship between diabetes and periodontal disease has been emphasized; additionally, metabolome analysis based on MS, which has further advanced, will continue to be an essential tool to elucidate the mechanism.

New treatments may be developed by elucidating the function of proteins that serve as diagnostic markers for the inflammation of periodontal tissues to control periodontopathogenic bacteria, which may further contribute to periodontal disease. Studying periodontal metabolite biomarkers involves several challenges. Therefore, when initiating research on metabolite biomarkers, it is essential to examine how specimens should be collected and analyzed and what evaluation method should be used. These questions should be addressed as the use of metabolomics increases in periodontology and other research fields.

## 5. Future Directions

The oral microbiome is the focus of increasing attention in periodontal research. The results of an analysis comparing the oral microbiomes at multiple sites (saliva and dental plaque) in a sample of 1000 people from the Japanese population were recently published [127]. The analysis revealed “differences in community structure between the microbiomes” of the saliva and dental plaque and showed that microbial diversity correlated with the severity of periodontal diseases [127]. Moreover, numerous studies on periodontal diseases have combined the microbiome and metabolome to elucidate the pathogenesis of periodontal diseases, such as the functional diversity of the microbial community in healthy participants and patients with periodontitis based on sole carbon source utilization by Zhang et al. [128]. Pei et al. reported a microbial and metabolomic analysis of GCF in patients with chronic periodontitis in relation to lessons for a predictive, preventive, and personalized medical approach [129]. In a cross-sectional observational study, Na et al. reported a molecular subgroup of periodontitis through an integrated analysis of the microbiome and metabolome [130].

The researchers report that they will make the results of their microbiome analysis widely available on the Internet without personal identifiers, and detailed results will be distributed for use by researchers nationwide. In the future, MS-based oral microbiome and metabolome analyses may be used to elucidate the pathogenesis of periodontal diseases.

## 6. Conclusions

It is expected that new treatments and drugs that can control the function of proteins and metabolites will be found. Biomolecules associated with inflammation, immune response, and tissue destruction in periodontal disease are expected to be valuable biomarkers for assessing periodontal disease activity and the response to diagnostic treatment. GCFs collected from the gingival sulcus and periodontal pockets contain biomarkers that reflect inflammation, immune response, and tissue destruction at the site of periodontal lesions, making metabolomic analysis using MS an essential tool for evaluation and diagnosis. Additionally, it is hoped that the analysis of many protein metabolites will clarify the functional links between metabolites whose expression fluctuates in relation to diseases, drugs and other protein metabolites and elucidate the mechanisms of periodontal disease development.

## Figures and Tables

**Figure 1 molecules-27-02864-f001:**
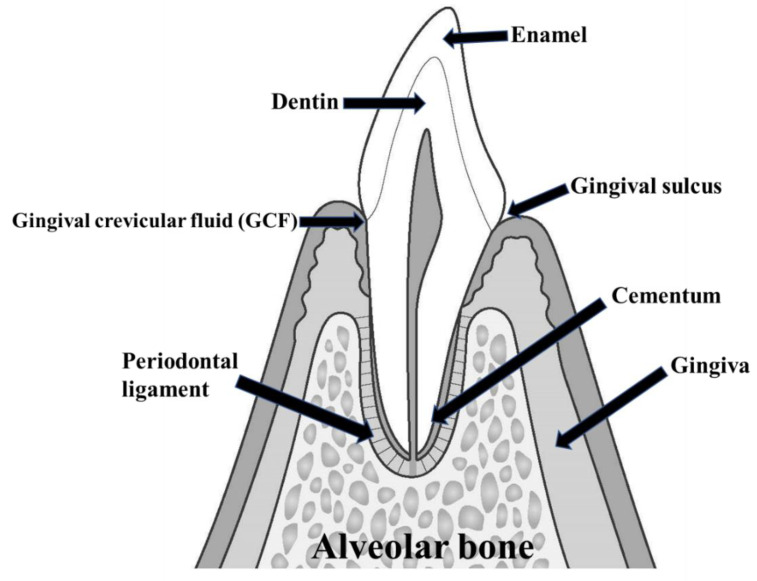
Periodontal tissue and other important factors. Periodontal tissue refers to the tissue that surrounds the teeth and supports their function. Periodontal tissue comprises cementum, gingiva, alveolar bone, and periodontal ligament. Periodontal disease is a general term for diseases that develop in periodontal supporting tissue. GCF, which is the fluid that exudes into the gingival sulcus and periodontal pockets, is thought to reflect the pathology of periodontal disease.

**Figure 2 molecules-27-02864-f002:**
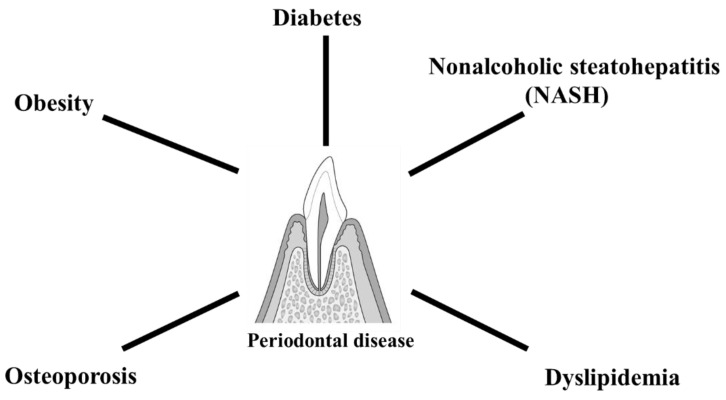
Relationship between periodontal disease and metabolic diseases. Metabolic disease groups that have been reported to be associated with periodontal disease include diabetes, obesity, nonalcoholic steatohepatitis (NASH), dyslipidemia, and osteoporosis.

**Figure 3 molecules-27-02864-f003:**
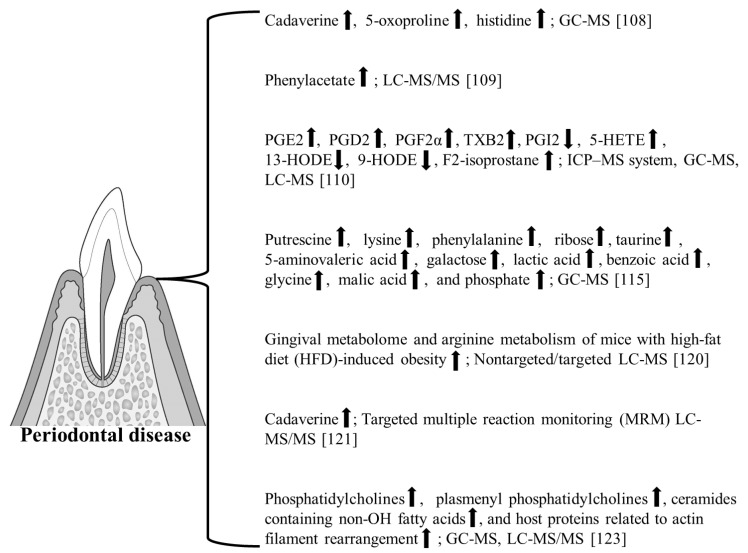
We suggest that the above are involved in periodontal disease. Summary of important findings of these studies.

**Table 1 molecules-27-02864-t001:** Characteristics of typical mass spectrometry methods.

Common Instrument Names	Overview	Analysis Target	Feature	Preprocessing	Applications-Clinical Laboratory [Ref]
GC-MS; GC-MS/MS	Consists of two instruments with distinct separation methods: a gas chromatograph (GC) for chromatographic separation and a mass spectrometer (MS) for mass separation	**Gas** Thermostable volatile substances or those that can be converted to thermostable volatile substances by derivatization. Used to estimate organic acids and steroids.	〇 Effective for the analysis of relatively small molecular weight and high-volatility compounds. 〇 Capable of qualitatively and quantitatively analyzing trace amounts of organic compounds. 〇 Various introduction methods can be chosen according to sample conditions and objectives. 〇 Extensive libraries enable the structural estimation of a wide range of organic compounds.	Liquid–liquid extraction. This technique is rather complicated.	〇 Metabolome analysis [55,56,57,58,59] 〇 Pharmaceutical and toxicological analysis [60,61,62,63]
LC-MS; LC-MS/MS	LC-MS/MS stands for Liquid Chromatograph—Mass Spectrometry, which combines high-performance liquid chromatography (HPLC) and triple quadrupole mass spectrometry (MS/MS). By combining the two methods, it separates the organic compounds in a liquid and analyzes them by mass.	**Liquid** Available for both fat- and water-soluble compounds and low to high molecular weight molecules.Drugs, catecholamine metabolites, fatty acid fractions, vitamin B1, carnitine, mucopolysaccharides, etc. can be estimated.	〇 Identification and quantification of non-volatile organic components. 〇 Identification and quantification of pyrolysable organic components. 〇 Determination of the molecular structure of unknown organic components (*should be utilized in combination with other instruments).	Pretreatment with solid-phase columns. The technique is relatively simple.	〇 Screening of congenital metabolic disorders [64,65,66,67,68] 〇 Method toxicology, TDM [69,70,71,72] 〇 Clinical chemistry (small molecules, amino acids, peptides, and proteins) [73,74,75,76] 〇 Genotyping [77,78,79]
MALDI-TOF MS	Matrix-Assisted Laser Desorption/Ionization-Time-of-Flight type mass spectrometer (MALDI-TOF MS): This is a type of mass spectrometer that combines MALDI as the ionization method and TOF as the analyzer.	**Solid** As there is no separation required prior to ionization, it can be targeted in solid form without dissolution or homogenization.	〇 High resolution and accurate mass estimation. 〇 Ionization of high molecular weight samples is possible. 〇 Primarily, monovalent ions are observed in the mass spectrum. 〇 High sensitivity and measurement is possible with a small sample (a few μL).	Matrix application. The technique is simple.	〇 Rapid identification of bacteria and fungi [80,81,82,83] 〇 Mass spectrometry imaging [84,85,86,87,88]

**Table 2 molecules-27-02864-t002:** Summary of important findings of these studies.

Authors	Mass Spectrometer Used in the Analysis	The Following Are Those Suggested to Be Involved with Periodontal Disease	References
Kuboniwa et al.	GC-MS	Cadaverine, 5-oxoproline, histidine	[108]
Liebsch et al.	LC-MS/MS	Phenylacetae	[109]
Huang et al.	ICP–MS system, GC-MS, LC-MS	PGE2, PGD2, PGF2α, TXB2, PGI2, 5-HETE, 13-HODE, 9-HODE, F2-isoprostane	[110]
Ozaki et al.	GC-MS	Putrescine, lysine, phenylalanine, ribose, taurine, 5-aminovaleric acid, and galactose, Lactic acid, benzoic acid, glycine, malic acid, phosphate	[115]
Chen et al.	Nontargeted/targeted LC-MS	Gingival metabolome and arginine metabolism of mice with high-fat diet (HFD)-induced obesity	[120]
Schulte et al.	Targeted multiple reaction monitoring (MRM) LC-MS/MS	Cadaverine	[121]
Overmyer et al.	GC-MS, LC-MS/MS	Phosphatidylcholines, plasmenyl phosphatidylcholines, ceramides containing non-OH fatty acids, host proteins related to actin filament rearrangement	[123]

## Data Availability

Not applicable.

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
