# Peer review of "Metabolomics Research in Periodontal Disease by Mass Spectrometry"

_molecules, 2022, doi:10.3390/molecules27092864_

Round 1

Reviewer 1 Report

The article title is very interesting. The author should have more efforts to piece the information what readers are expected. I have some comments for author that are improved the article:

  1. Table 1 and 2 look the same and lack of information
  2. Table 3 having lack of information, author should provided more information about regulation of metabolites in different studies.
  3. Quality of artwork is very poor. It should be improved.

Reviewer 2 Report

Comments on "Metabolomics Research in Periodontal Disease by Mass Spectrometry" by Tsuchida and Nakayama.

The manuscript is well written and has covered the potential areas. Authors have enlisted the proteins, metal ions and metabolites levels found in periodontal diseases and their association with other metabolic syndromes. It will be nice if authors could put together this textual information in a model as Fig. 3.  and prepare a model based on different metabolites, metal ions and proteins that were shown altered under disease conditions, it would be of great importance to put together finding of these studies into a concise model for the benefit of the readers. Review should end with a model.  It will certainly benefit the manuscript.

Reviewer 3 Report

"Metabolomics Research in Periodontal Disease by Mass Spectrometry is a good review" on periodontology (a newer field relative to other areas of dentistry) at the right time considering the rapid progress of metabolomics analysis in various research fields and in discovering novel markers and diagnostic biomarker candidates for various diseases using different MS technologies.

The author has provided a good introduction on periodontal disease (with a good figure), metabolomics, and mass spectrometry including citing a good number of literature. The author then detailed very well on periodontal disease and its relationship with different metabolic diseases, and then detailed on mass spectrometry of metabolomics analysis by briefly highlighting the use of MS technology in clinical laboratories in a table format and citing respective reference literature.

Later the author discussed on the applications of metabolomics research in periodontal disease using MS by giving 7 literature examples (although may be a few more examples would have been better) along with a good table format to easily visualize these applications to the audience. Also, concluded well with future directions.

Overall the author presented this review very well and would be very helpful to the researchers who are currently working on metabolomics-and-periodontal diseases and this review gives good insights on the importance of using metabolomics analysis and MS-based studies (for eg., oral microbiome) that eventually can help in elucidating the pathogenesis and mechanisms of periodontal disease development, and also other new treatments.

Author Response

Please see the attachemnet.

Round 2

Reviewer 1 Report

Thank you so much for revising the manuscript and updating it